# Qimera: Data-free Quantization with Synthetic Boundary Supporting Samples

**Kanghyun Choi**[1]     **Deokki Hong**[2]     **Noseong Park**[1,2]     **Youngsok Kim**[1,2]     **Jinho Lee**[1,2]*

[1] Department of Computer Science, Yonsei University
[2] Department of Artificial Intelligence, Yonsei University
{kanghyun.choi, dk.hong, noseong, youngsok, leejinho}@yonsei.ac.kr

## Abstract

Model quantization is known as a promising method to compress deep neural networks, especially for inferences on lightweight mobile or edge devices. However, model quantization usually requires access to the original training data to maintain the accuracy of the full-precision models, which is often infeasible in real-world scenarios for security and privacy issues. A popular approach to perform quantization without access to the original data is to use synthetically generated samples, based on batch-normalization statistics or adversarial learning. However, the drawback of such approaches is that they primarily rely on random noise input to the generator to attain diversity of the synthetic samples. We find that this is often insufficient to capture the distribution of the original data, especially around the decision boundaries. To this end, we propose Qimera, a method that uses superposed latent embeddings to generate synthetic boundary supporting samples. For the superposed embeddings to better reflect the original distribution, we also propose using an additional disentanglement mapping layer and extracting information from the full-precision model. The experimental results show that Qimera achieves state-of-the-art performances for various settings on data-free quantization. Code is available at https://github.com/iamkanghyunchoi/qimera.

## 1   Introduction

Among many neural network compression methodologies, quantization is considered a promising direction because it can be easily supported by accelerator hardwares [1] than pruning [2] and is more lightweight than knowledge distillation [3]. However, quantization generally requires some form of adjustment (e.g., fine-tuning) using the original training data [4, 5, 6, 7, 8] to restore the accuracy drop due to the quantization errors. Unfortunately, access to the original training data is not always possible, especially for deployment in the field, for many reasons such as privacy and security. For example, the data could be medical images of patients, photos of confidential products, or pictures of military assets.

Therefore, data-free quantization is a natural direction to achieve a highly accurate quantized model without accessing any training data. Among many excellent prior studies [9, 10, 11, 12], generative methods [13, 14, 15] have recently been drawing much attention due to their superior performance. Generative methods successfully generate synthetic samples that resemble the distribution of the original dataset and achieve high accuracy using information from the pretrained full-precision network, such as batch-normalization statistics [15, 13] or intermediate features [14].

---

*Corresponding author

35th Conference on Neural Information Processing Systems (NeurIPS 2021).

However, a significant gap still exists between data-free quantized models and quantized models fine-tuned with original data. What is missing from the current generative data-free quantization schemes? We hypothesize that the synthetic samples of conventional methods lack *boundary supporting samples* [16], which lie on or near the decision boundary of the full-precision model and directly affect the model performance. The generator designs are often based on conditional generative adversarial networks (CGANs) [17, 18] that take class embeddings representing class-specific latent features. Based on these embeddings as the centroid of each class distribution, generators rely on the input of random Gaussian noise vectors to gain diverse samples. However, one can easily deduce that random noises have difficulty reflecting the complex class boundaries. In addition, the weights and embeddings of the generators are trained with cross-entropy (CE) loss, further ensuring that these samples are well-separated from each other.

In this work, we propose Qimera, a method for data-free quantization employing superposed latent embeddings to create boundary supporting samples. First, we conduct a motivational experiment to confirm our hypothesis that samples near the boundary can improve the quantized model performance. Then, we propose a novel method based on inputting superposed latent embeddings into the generator to produce synthetic boundary supporting samples. In addition, we provide two auxiliary schemes for flattening the latent embedding space so that superposed embeddings could contain adequate features.

Qimera achieves significant performance improvement over the existing techniques. The experimental results indicate that Qimera sets new state-of-the-art performance for various datasets and model settings for the data-free quantization problem. Our contributions are summarized as the following:

- We identify that boundary supporting samples form an important missing piece of the current state-of-the-art data-free compression.

- We propose using superposed latent embeddings, which enables a generator to synthesize boundary supporting samples of the full-precision model.

- We propose disentanglement mapping and extracted embedding initialization that help train a better set of embeddings for the generator.

- We conduct an extensive set of experiments, showing that the proposed scheme outperforms the existing methods.

## 2   Related Work

### 2.1   Data-free Compression

Early work on data-free compression has been led by knowledge distillation [3], which usually involves pseudo-data created from teacher network statistics [19, 20]. Lopes et al. [19] suggested generating pseudo-data from metadata collected from the teacher in the form of activation records. Nayak et al. [20] proposed a similar scheme but with a zero-shot approach by modeling the output space of the teacher model as a Dirichlet distribution, which is taken from model weights. More recent studies have employed generator architectures similar to GAN [21] to generate synthetic samples replacing the original data [22, 23, 24, 25]. In the absence of the original data for training, DAFL [22] used the teacher model to replace the discriminator by encouraging the outputs to be close to a one-hot distribution and by maximizing the activation counts. KegNet [26] adopted a similar idea and used a low-rank decomposition to aid the compression. Adversarial belief matching [23] and data-free adversarial distillation [24] methods suggested adversarially training the generator, such that the generated samples become harder to train. One other variant is to modify samples directly using logit maximization as in DeepInversion [25]. While this approach can generate images that appear natural to a human, it has the drawback of skyrocketing computational costs because each image must be modified using backpropagation.

Data-free quantization is similar to data-free knowledge distillation but is a more complex problem because quantization errors must be recovered. The quantized model has the same architecture as the full-precision model; thus, the early methods of post-training quantization were focused on how to convert the full-precision weights into quantized weights by limiting the range of activations [9, 12], correcting biases [9, 10], and equalizing the weights [10, 12]. ZeroQ [15] pioneered using synthetic data for data-free quantization employing statistics stored in batch-normalization layers. GDFQ [13] added a generator close to the ACGAN [18] to generate better synthetic samples, and

| Data | Accuracy | Data-free |
|------|----------|-----------|
| Fine-tuned with Original data | 68.28% | ✗ |
| Synthetic samples | 63.39% | ✓ |
| Synthetic + unconfusing real samples | 63.91% (+0.52) | ✗ |
| Synthetic + confusing real samples | 65.75% (+2.36) | ✗ |
| Qimera (This work) | 65.10% (+1.71) | ✓ |

Table 1: Motivational Experiment

AutoReCon [27] suggests a better performing generator found by neural architecture search. In addition, DSG [28] further suggests relaxing the batch-normalization statistics alignment to generate more diverse samples. ZAQ [14] adopted adversarial training of the generator on the quantization problem and introduced intermediate feature matching between the full-precision and quantized model. However, none of these considered aiming to synthesize boundary supporting samples of the full-precision model. Adversarial training of generators have a similar purpose, but it is fundamentally different from generating boundary supporting samples. In addition, adversarial training can be sensitive to hyperparameters and risks generating samples outside of the original distributions [29]. To the extent of our knowledge, this work is the first to propose using superposed latent embeddings to generate boundary supporting samples explicitly for data-free quantizations.

## 2.2 Boundary Supporting Samples

In the context of knowledge distillation, boundary supporting samples [16] are defined as samples that lie near the decision boundary of the teacher models. As these samples contain classification information about the teacher models, they can help the student model correctly mimic the teacher's behavior. Heo et al. [16] applied an adversarial attack [30] to generate boundary supporting samples and successfully demonstrated that they improve knowledge distillation performance. In AMKD [31], triplet loss was used to aid the student in drawing a better boundary. Later, DeepDig [32] devised a refined method for generating boundary supporting samples and analyzed their characteristics by defining new metrics. Although boundary supporting samples have been successful for many problems, such as domain adaptation [33] and open-set recognition [34], they have not yet been considered for data-free compression.

## 3 Motivational Experiment

To explain the discrepancy between the accuracy of the model fine-tuned with the original training data and the data-free quantized models, we hypothesized that the synthetic samples from generative data-free quantization methods lack samples near the decision boundary. To validate this hypothesis, we designed an experiment using the CIFAR-100 [35] dataset with the ResNet-20 network [36].

First, we forwarded images in the dataset into the pre-trained full-precision ResNet-20. Among these, we selected 1500 samples (3% of the dataset, 15 samples per class) from samples where the highest confidence value was lower than 0.25, forming a group of 'confusing' samples. Then, we combined the confusing samples with synthetic samples generated from a previous study [13] to fine-tune the quantized model with 4-bit weights and 4-bit activations. We also selected as a control group an equal number of random samples from the images classified as unconfusing and fine-tuned the quantized model using the same method.

The results are presented in Table 1. The quantized model with synthetic + unconfusing real samples exhibited only 0.52%p increase in the accuracy from the baseline. In contrast, adding confusing samples provided 2.36%p improvement, filling almost half the gap towards the quantized model fine-tuned with the original data. These results indirectly validate that the synthetic samples suffer from a lack of confusing samples (i.e., boundary supporting samples). We aim to address the issue in this paper. As we indicate in Section 5, Qimera achieves a 1.71%p performance gain for the same model in a data-free setting, close to that of the addition of confusing synthetic samples.

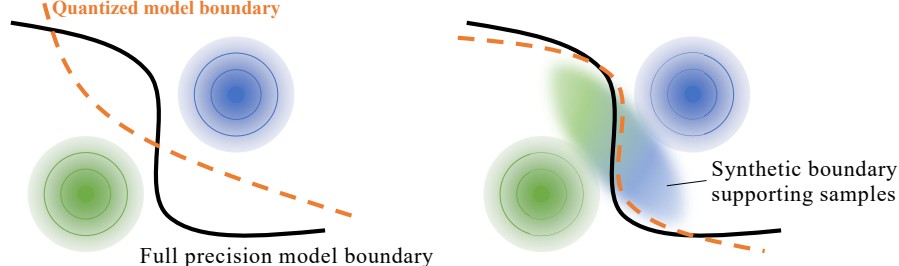

(a) Synthetic Images generated
with Gaussian noise.

(b) Synthetic boundary supporting samples with superposed latent embeddings (proposed).

Figure 1: Diagram of synthetic samples in the feature space of the full-precision network. The black curves represent the decision boundary of the full-precision model, which are considered ideal for the quantized model (orange dotted curves) to mimic. When synthetic images are generated with per-class embeddings and noises as in (a), their features do not support the decision boundary, whereas the proposed approach in (b) generates the boundary supporting samples, helping the quantized model to set the near-ideal decision boundary.

## 4 Generating Boundary Supporting Samples with Superposed Latent Embeddings

### 4.1 Baseline Generative Data-free Quantization

Recent generative data-free quantization schemes [13, 14] employ a GAN-like generator to create synthetic samples. In the absence of the original training samples, the generator $G$ attempts to generate synthetic samples so that the quantized model $Q$ can mimic the behavior of the full-precision model $P$. For example, in GDFQ [13], the loss function $\mathcal{L}_{GDFQ}$ is

$$\mathcal{L}_{GDFQ}(G) = \mathcal{L}_{CE}^P(G) + \alpha\mathcal{L}_{BNS}^P(G), \tag{1}$$

where the first term $\mathcal{L}_{CE}$ guides the generator to output clearly classifiable samples, and the second term $\mathcal{L}_{BNS}$ aligns the batch-normalization statistics of the synthetic samples with those of the batch-normalization layers in the full-precision model. In another previous work ZAQ [14],

$$\mathcal{L}_{ZAQ}(G) = \mathcal{L}_o^{P,Q}(G) + \beta\mathcal{L}_f^{P,Q}(G) + \gamma\mathcal{L}_a^P(G), \tag{2}$$

where the first term $\mathcal{L}_o$ separates the prediction outputs of $P$ and $Q$, and the second term $\mathcal{L}_f$ separates the feature maps produced by $P$ and $Q$. These two losses let the generator be adversarially trained and allow it to determine samples where $P$ cannot mimic $Q$ adequately. Lastly, the third term $\mathcal{L}_a$ maximizes the activation map values of $P$ so that the created samples do not drift too far away from the original dataset.

The quantized model $Q$ is usually jointly trained with $G$, such that

$$\mathcal{L}_{GDFQ}(Q) = \mathcal{L}_{CE}^Q(Q) + \delta\mathcal{L}_{KD}^P(Q), \tag{3}$$

for GDFQ, and

$$\mathcal{L}_{ZAQ}(Q) = -\mathcal{L}_o^{P,Q}(G) - \beta\mathcal{L}_f^{P,Q}(G), \tag{4}$$

for ZAQ, respectively, where $\mathcal{L}_{KD}$ from Eq. 3 is the usual KD loss with Kullback-Leibler divergence.

While these two methods exhibit great performance, they both model the distribution of per-class embeddings in the latent input space as a Gaussian distribution, and generate diverse samples using random Gaussian noise inputs. However, based on the Gaussian distribution, it is difficult to correctly reflect the boundary between the two different classes, especially when all samples have one-hot labels. Figure 1 visualizes such problems in a simplified two-dimensional space. With samples generated from gaussian noise (Figure 1a), the two per-class distributions are far away, and a void exists between the two classes (see Figure 3b for plots from experimental data). This can cause the decision boundaries to be formed at nonideal regions. One alternative is to use noise with higher

Figure 2: An overview of the proposed method. Proposed components are denoted as colored shapes.

variance, but the samples would overlap on a large region, resulting in too many samples with incorrect labels.

Furthermore, in the above approaches, the loss terms of Eqs. 1 and 2 such as $\mathcal{L}_{CE}$ and $\mathcal{L}_o$ encourage the class distributions to be separated from each other. While this is beneficial for generating clean, well-classified samples, it also prevents generating samples near the decision boundary, which is necessary for training a good model [16, 31]. Therefore, in this paper, we focus on methods to generate synthetic samples near the boundary from the full-precision model $P$ (Figure 1b).

## 4.2   Superposed Latent Embeddings

The overview of Qimera is presented in Figure 2. Often, generators use learned embeddings to create samples of multiple classes [18, 37]. Given an input one-hot label $y$ representing one of $C$ classes and a random noise vector $\mathbf{z}$, a generator $G$ uses an embedding layer $E \in \mathbb{R}^{D \times C}$ to create a synthetic sample $\hat{x}$:

$$\hat{x} = G(\mathbf{z} + E_y), \qquad\qquad \mathbf{z} \sim \mathcal{N}(0, 1). \qquad (5)$$

To create boundary supporting samples, we superpose the class embeddings so that the generated samples have features lying near the decision boundaries of $P$. With two embeddings superposed, new synthetic sample $\hat{x}'$ becomes

$$\hat{x}' = G(\mathbf{z} + (\lambda E_{y_1} + (1 - \lambda)E_{y_2})), \qquad \mathbf{z} \sim \mathcal{N}(0, 1),\ 0 \le \lambda \le 1. \qquad (6)$$

To avoid too many confusing samples from complicating the feature space, we also apply soft labels in the same manner as a regularizer:

$$\hat{y}' = \lambda y_1 + (1 - \lambda)y_2, \qquad\qquad 0 \le \lambda \le 1. \qquad (7)$$

Generalizing to $K$ embeddings, Eqs. 6 and 7 become

$$(\hat{x}', \hat{y}') = \Big(G\big(S(e)\big), \sum_k^K \lambda_k y_k\Big),\ \ S(e) = \mathbf{z} + \sum_k^K \lambda_k e_k, \qquad \mathbf{z} \sim \mathcal{N}(0, 1), \qquad (8)$$

$$\lambda_i = Softmax(p_i) = exp(p_i)/\textstyle\sum_k^K (exp(p_k)), \qquad p_i \sim \mathcal{N}(0, 1). \qquad (9)$$

where $e \in \mathbb{R}^{D \times K}$ has $K$ embeddings from $E$ as the column vectors (i.e., $e = [E_{y_0}, \ldots, E_{y_{K-1}}]$), and $S : \mathbb{R}^{D \times K} \to \mathbb{R}^D$ is a superposer function. Applying this to existing methods is straightforward and incurs only a small amount of computational overhead. Similar to knowledge distillation with boundary supporting samples [16, 31], the superposed embeddings are supposed to help transfer the decision boundary of the full precision (teacher) model to the quantized (student) model.

Although the superposed embedding scheme alone produces a substantial amount of performance gains, the generator embedding space is often not flat enough [38]; therefore, linearly interpolating them can result in unnatural samples [39]. To mitigate this, the embedding space used in Qimera should possess two characteristics. First, the embedding space should be as flat as possible so that the samples generated from Eq. 8 reflect the intermediate points in the feature space. Second, the individual embeddings should still be sufficiently distinct from each other, correctly representing the distance between each class distribution. In the remaining two subsections, we describe our strategies for enforcing the embeddings to contain the above characteristics.

### 4.3 Disentanglement Mapping

To perform superposing in a flatter manifold, we added a learnable mapping function $M : \mathbb{R}^D \rightarrow \mathbb{R}^d$ before the embeddings are superposed, where $D$ is the embedding dimension and $d$ is the dimension of the target space. Thus, $\hat{x}'$ from Eq. 8 becomes $\hat{x}' = G\big(S(m)\big)$, where $m_k = M(e_k)$. Although we do not add any specific loss that guides the output space of $M$ to be flat, training to match the output of the full-precision model (i.e., $\mathcal{L}_{CE}$) using the superposed $\hat{y}'$ encourages $M$ to map the input to a flatter space. In practice, we modeled $M$ as a single-layer perceptron, which we call the disentanglement mapping layer. The experimental results from Section 5 demonstrate that the disentanglement mapping provides a considerable amount of performance gain.

### 4.4 Extracted Embedding Initialization

For the embeddings to be flat, we want the distributions of the embeddings fused with noise to be similar to the feature space. For this purpose, we utilize the fully connected layer of the full precision model $P$. Given $f$, the output features of the full-precision model before the last fully connected layer, we minimize $\sum_y^C dist\big(P(f|y), P(g|y)\big)$, where $C$ is the number of classes, $g$ is the input of the generator, and $dist$ is some distance metric. We do not have knowledge of the distribution of $f$; thus, we modeled it as a Gaussian distribution, such that $f|y \sim \mathcal{N}(\mu_y, 1)$. Therefore, solving it against Eq. 5 simply yields $E(y) = \mu_y$. In practice, we use the corresponding column from the weight of the last fully connected layer of the full-precision model because its weights represent the centroids of the activations. If the weights of the last fully connected layer $\mathbf{W} = [\mathbf{w_1}, \mathbf{w_2}, ..., \mathbf{w_C}]$, we set $\mu_{\mathbf{y}} = \mathbf{w_y}$. Our experiments reveal that extracting the weights from the full-precision model and freezing them already works well (see Appendix B.2). However, using them as initializations and jointly training them empirically works better. We believe this is because fully connected layers have bias parameters in addition to weight parameters. Because we do not extract these biases into the embeddings, a slight tuning is needed by training them. This outcome aligns with the findings from the class prototype scheme used in self-supervised learning approaches [40, 33].

## 5 Experimental Results

### 5.1 Experiment Implementation

Our method is evaluated on CIFAR-10, CIFAR-100 [35] and ImageNet (ILSVRC2012 [41]) datasets, which are well-known datasets for evaluating the performance of a model on the image classification task. CIFAR10/100 datasets consist of 50k training sets and 10k evaluation sets with 10 classes and 100 classes, respectively, and is used for small-scale experiments in our evaluation. ImageNet dataset is used for large-scale experiments. It has 1.2 million training sets and 50k evaluation sets. To keep the data-free environment, only the evaluation sets were used for test purposes in all experiments.

For the experiments, we chose ResNet-20 [36] for CIFAR-10/100, and ResNet-18, ResNet-50, and MobileNetV2 [42] for ImageNet. We implemented all our techniques using PyTorch [43] and ran the experiments using RTX3090 GPUs. All the model implementations and pre-trained weights before quantization are from pytorchcv library [44]. For quantization, we quantized all the layers and activations using $n$-bit linear quantization, described by [7], as below:

$$\theta' = \left\lfloor \frac{\theta - \theta_{min}}{interval(n)} - 2^{n-1} \right\rceil \tag{10}$$

where $\theta$ is the full-precision value, $\theta'$ is the quantized value, $interval(n)$ is calculated as $\frac{\theta_{max} - \theta_{min}}{2^n - 1}$. $\theta_{min}$ and $\theta_{max}$ are per-channel minimum and maximum value of $\theta$.

To generate synthetic samples, we built a generator using the architecture of ACGAN [18] and added a disentanglement mapping layer after class embeddings followed by a superposing layer. Among all batches, the superposing layer chooses between superposed embeddings and regular embeddings in $p : 1 - p$ ratio. The dimension of latent embedding and random noise was set to be the same with the channel of the last fully connected layer of the target network. For CIFAR, the intermediate embedding dimension after the disentanglement mapping layer was set as 64. The generator was trained using Adam [45] with a learning rate of 0.001. For ImageNet, the intermediate embedding dimension was set to be 100. To maintain label information among all layers of the generator, we

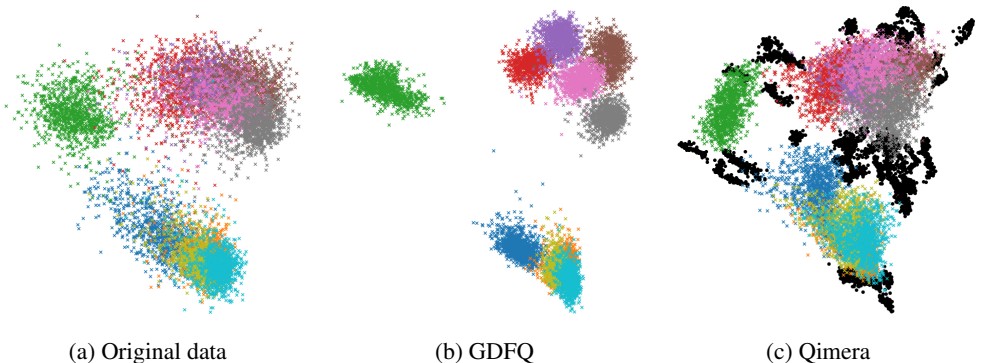

|                 (a) Original data                 |                 (b) GDFQ                 |                 (c) Qimera                 |

Figure 3: PCA plots of the features before the last layer of the full-precision model. In the plots of Qimera-generated samples (c), the black dots represent synthetic boundary supporting samples generated with the superposed latent embeddings, which fill the space between the clusters. GDFQ-generated samples (b) form clusters that are smaller than that of the original data (a) and lack samples in the mid-ground. Best viewed in color.

apply conditional batch normalization [46] rather than regular batch normalization layer, following SN-GAN [47]. The optimizer and learning rate were the same as that of CIFAR.

To fine-tune the quantized model $Q$, we used SGD with Nesterov [48] as an optimizer for $Q$ with a learning rate of 0.0001 while momentum and weight decay terms as 0.9 and 0.0001 respectively. The generator $G$ and the quantized model $Q$ were jointly trained with 200 iterations for 400 epochs while decaying the learning rate by 0.1 per every 100 epochs. The batch size was 64 and 16 for CIFAR and ImageNet respectively. While Qimera can be applied almost all generator-based approaches, we chose to adopt baseline loss functions for training from GDFQ [13], because it was stable and showed better results for large scale experiments. Thus, loss functions $\mathcal{L}(G)$ and $\mathcal{L}(Q)$ are equal to Eq. 1 and Eq. 3 with $\alpha = 0.1$ and $\delta = 1.0$, following the baseline.

## 5.2 Visualizations of Qimera-generated Samples on Feature Space

To ensure that the superposed latent embedding creates boundary supporting samples, we conducted an experiment to compare generated synthetic samples on feature space visually. The experimental results are based on the generators trained with ResNet-20 for CIFAR-10 dataset. The features were extracted from the intermediate activation before the last fully connected layer of the full-precision model. For the Qimera-based generator, we set K=2 during the sample generations for clarity.

After extracting the features, we projected the features into a two-dimensional space using Principal Component Analysis (PCA) [49]. The results are presented in Figure 3. Compared with the original training set data (Figure 3a), the samples from the GDFQ-based generator (Figure 3b) show a lack of boundary supporting samples and the class distributions are confined to small regions around the centroid of each class. A generator trained with Qimera, however, exhibits different characteristics (Figure 3c). Samples that are generated from superposed latent embeddings are displayed as black dots. The black dots are mostly located on the sparse regions between the class clusters. This experimental result shows that our method, Qimera, successfully generates samples near the decision boundaries. In other words, superposed latent embeddings are not only superposed on embedding space but also in the feature space, serving as synthetic boundary supporting samples.

## 5.3 Quantization Results Comparison

Table 2 displays the classification accuracy on various datasets, target models, and bit-width settings. Note that $n$wm$a$ means $n$-bit quantization for weights and $m$-bit quantization for activations. As baselines, we selected ZeroQ [15], ZAQ [14], and GDFQ [13] as the important previous works on generative data-free quantization. In addition, we implemented Mixup [50] and Cutmix [51] on top of GDFQ, which are data augmentation schemes that mix input images. To implement these schemes, we created synthetic images from GDFQ, and applied the augmentations to build training images and labels. All baseline results are from official code of the authors, where a small amount of

| Dataset | Model (FP32 Acc.) | Bits | ZeroQ | ZAQ | GDFQ | GDFQ +Cutmix | GDFQ +Mixup | Qimera (%p improvement) |
|---|---|---|---|---|---|---|---|---|
| Cifar-10 | ResNet-20 (93.89) | 4w4a | 79.30 | **92.13**$^*$ | 90.25 | 89.58 | 88.69 (-3.44) | 91.26 $\pm$0.49 (-0.87) |
| | | 5w5a | 91.34 | 93.36 | 93.38$^*$ | 92.75 | 92.79 (-0.59) | **93.46 $\pm$0.03 (+0.08)** |
| Cifar-100 | ResNet-20 (70.33) | 4w4a | 47.45 | 60.42 | 63.39$^*$ | 62.74 | 62.99 (-0.40) | **65.10 $\pm$0.33 (+1.71)** |
| | | 5w5a | 65.61 | 68.70$^*$ | 66.12 | 67.51 | 67.78 (-0.92) | **69.02 $\pm$0.22 (+0.32)** |
| ImageNet | ResNet-18 (71.47) | 4w4a | 22.58 | 52.64 | 60.60$^*$ | 58.90 | 61.72 (+1.12) | **63.84 $\pm$0.30 (+3.24)** |
| | | 5w5a | 59.26 | 64.54 | 68.40$^*$ | 68.05 | 68.67 (+0.27) | **69.29 $\pm$0.16 (+0.89)** |
| | ResNet-50 (77.73) | 4w4a | 8.38 | 53.02$^*$ | 52.12 | 51.80 | 59.25 (+6.24) | **66.25 $\pm$0.90 (+13.23)** |
| | | 5w5a | 48.12 | 73.38$^*$ | 71.89 | 70.99 | 71.57 (-1.81) | **75.32 $\pm$0.09 (+1.94)** |
| | MobileNetV2 (73.03) | 4w4a | 10.96 | 0.10$^\dagger$ | 59.43$^*$ | 57.23 | 59.99 (+0.56) | **61.62 $\pm$0.39 (+2.19)** |
| | | 5w5a | 59.88 | 62.35 | 68.11$^*$ | 67.61 | 68.83 (+0.72) | **70.45 $\pm$0.07 (+2.34)** |

$^*$ Highest among the baselines  $^\dagger$Did not converge

Table 2: Comparison on data-free quantization schemes

| Dataset | Model (FP32 Acc.) | Bits | GDFQ | Qimera + GDFQ | ZAQ | Qimera + ZAQ | AutoReCon | Qimera + AutoReCon |
|---|---|---|---|---|---|---|---|---|
| Cifar-10 | ResNet-20 (93.89) | 4w4a | 90.25 | 91.26 (+1.01) | 92.13 | **93.91 (+1.78)** | 88.55 | 91.16 (+2.61) |
| | | 5w5a | 93.38 | 93.46 (+0.08) | 93.36 | **93.84 (+0.48)** | 92.88 | 93.42 (+0.54) |
| Cifar-100 | ResNet-20 (70.33) | 4w4a | 63.39 | 65.10 (+1.71) | 60.42 | **69.30 (+8.88)** | 62.76 | 65.33 (+2.57) |
| | | 5w5a | 66.12 | 69.02 (+2.90) | 68.70 | **69.58 (+0.88)** | 68.40 | 68.80 (+0.40) |

Table 3: Performance of Qimera implemented on top of GDFQ, ZAQ and AutoReCon

modifications have been made on the GDFQ baseline for applying Mixup and Cutmix. We report top-1 accuracy for each experiment. The numbers inside the parentheses of Qimera results are improvements over the highest baseline.

The results demonstrate that Qimera outperforms the baselines at almost all settings. Its performance improvement is especially large for low-bitwidth (4w4a) cases. For 5w5a setting, the gain is still significant, and its performance reaches near that of the full-precision model, which represents the upper bound. In addition, Qimera is not limited to small datasets having a low spatial dimension. The ImageNet results prove that Qimera performs beyond other baselines with considerable gaps, on a large-scale dataset with many classes. Interestingly, the result of GDFQ+Mixup and GDFQ+Cutmix did not produce much noticeable improvement except for GDFQ+Mixup in 4w4a ResNet -18 and -50. This implies that a mixture of generated synthetic samples of different classes in the sample space is not sufficient to represent boundary supporting samples. In summary, Qimera achieves superior accuracy on various environments regardless of dataset or model scale.

### 5.4 Qimera on Top of Various Algorithms

In this paper, we have applied GDFQ [13] as a baseline. However, our design does not particularly depend on a certain method, and can be adopted by many schemes. To demonstrate this, we implemented Qimera on top of ZAQ [14] and AutoReCon [27]. The results are shown in Table 3.

AutoReCon [27] strengthens the generator architecture using a neural architecture search, and therefore applying this is no different from that of the original Qimera implemented on top of GDFQ. However, original ZAQ does not use per-class sample generation. To attach the techniques from Qimera, we extended the generator with an embedding layer, and added a cross-entropy loss into the loss function as the following:

$$\mathcal{L}(G) = \mathcal{L}_o^{P,Q}(G) + \beta\mathcal{L}_f^{P,Q}(G) + \gamma\mathcal{L}_a^P(G) + \rho\mathcal{L}_{CE}^P(G), \tag{11}$$

$$\mathcal{L}_{(Q)} = -\mathcal{L}_o^{P,Q}(G) - \beta\mathcal{L}_f^{P,Q}(G) + \rho\mathcal{L}_{CE}^Q(G), \tag{12}$$

where $\rho$ was set to 0.1, and $\mathcal{L}_{CE}^P(G)$, $\mathcal{L}_{CE}^Q(G)$ are calculated based on Eq. 8.

For all cases, applying the techniques of Qimera improves the performance by a significant amount. Especially for Qimera + ZAQ, the performances obtained was better than those of our primary implementation Qimera + GDFQ. Unfortunately, it did not converge with ImageNet, and thus we did not consider this implementation as the primary version of Qimera.

| Dataset | Method | $K$ | 2 | | 10 | | 25 | | 100 | | Best |
|---|---|---|---|---|---|---|---|---|---|---|---|
| | | $p$ | 0.4 | 0.7 | 0.4 | 0.7 | 0.4 | 0.7 | 0.4 | 0.7 | |
| Cifar-100 | Baseline | | N/A | N/A | N/A | N/A | N/A | N/A | N/A | N/A | 63.39 |
| (ResNet-20) | +SE[1] | | 64.14 | 64.55 | **64.55** | 64.37 | 64.21 | 64.11 | 63.77 | 64.26 | 64.55 (+1.16) |
| | +SE, DM[2] | | 64.73 | **64.76** | 64.01 | 64.22 | 63.60 | 64.32 | 63.82 | 64.63 | 64.76 (+1.37) |
| | +SE, EEI[3] | | 64.64 | 64.54 | 64.89 | 64.83 | **65.23** | 64.33 | 64.88 | 64.70 | **65.23 (+1.84)** |
| | +SE, DM, EEI | | 64.90 | 64.76 | **65.10** | 64.52 | 64.72 | 64.4 | 64.63 | 64.79 | 65.10 (+1.71) |

| Dataset | Method | $K$ | 100 | | 250 | | 500 | | 1000 | | Best |
|---|---|---|---|---|---|---|---|---|---|---|---|
| | | $p$ | 0.4 | 0.7 | 0.4 | 0.7 | 0.4 | 0.7 | 0.4 | 0.7 | |
| ImageNet | Baseline | | N/A | N/A | N/A | N/A | N/A | N/A | N/A | N/A | 52.12 |
| (ResNet-50) | +SE | | 57.34 | 60.93 | 59.27 | 62.91 | 63.12 | **64.09** | 61.04 | 60.89 | 64.09 (+11.98) |
| | +SE, DM | | 59.59 | 64.12 | 63.97 | 64.87 | 62.16 | 64.72 | 65.43 | **66.06** | 66.06 (+13.94) |
| | +SE, EEI | | 56.52 | 62.23 | 61.24 | 61.08 | 54.45 | 62.87 | 54.84 | **64.44** | 64.44 (+12.32) |
| | +SE, DM, EEI | | 61.43 | 63.87 | 62.16 | 64.5 | 59.82 | **66.25** | 59.18 | 65.12 | **66.25 (+14.13)** |

[1]Superposed Embeddings [2]Disentanglement Mapping [3]Extracted Embedding Initialization

Table 4: Ablation study and Sensitivity analysis

| Method | Cifar-10 | | Cifar-100 | |
|---|---|---|---|---|
| | Dist. Ratio | Intrusion | Dist. Ratio | Intrusion |
| Mixup | 2.44 | 0.800 | 3.14 | 0.400 |
| SE Only | 1.58 | 0.00260 | 1.64 | 0.053 |
| SE+DM | 1.67 | 0.00073 | 1.59 | 0.044 |
| SE+DM+EEI | 1.57 | 0.00013 | 1.52 | 0.029 |

Table 5: Embedding Distance Ratio and Intrusion Score

## 5.5 Further Experiments

We discuss some aspects of our method in this section, which are effect of each scheme introduced in Section 4 on the accuracy of $Q$, a sensitivity study upon various $p$ and $K$ settings, and a closer investigation into the effect of disentanglement mapping (DM) and extracted embedding initialization (EEI) on the embedding space. All experiments in this subsection are under 4w4a setting.

**Ablation study.** We conducted an ablation study by adding the proposed components one by one on the GDFQ baseline. The results are presented in Table 4. As we can see, the superposed embedding (SE) alone brings a substantial amount of performance improvement for both Cifar-100 (1.16%p) and ImageNet (11.98%p). On Cifar-100, addition of EEI provides much additional gain of 0.68%p, while that of DM is marginal. ImageNet, on the other hand, DM provides 1.97%p additional gain, much larger than that of EEI. When both of DM and EEI are used together with SE, the additional improvement over the best among (+SE, DM) and (+SE, EEI) is relatively small. We believe this is because DM and EEI serve for the similar purposes. However, applying both techniques can achieve near-best performance regardless of the dataset.

**Sensitivity Analysis.** Table 4 also provides a sensitivity analysis for $K$ (number of classes for superposed embeddings) and $p$ (ratio of synthetic boundary supporting samples within the dataset). As shown in the table, both parameter clearly have an impact on the performance. For Cifar-100, the sweet spot values for $K$ and $p$ are both small, while those of the ImageNet were both large. We believe is because ImageNet has more classes. Because the number of class pairs (and the decision boundaries) is a quadratic function of number of classes, there needs more boundary supporting samples with more embeddings superposed to draw the correct border.

**Investigation into the Embedding Space.** To investigate how DM and EEI help shaping the embedding space friendly to SE, we conducted two additional experiments with ResNet-20 in Table 5. First, setting $K$=2 (between two classes), we measured the ratio of the perceptual distance and Euclidean distance between the two embeddings in the classifier's feature space. The perceptual distance is measured by sweeping $\lambda$ from Eqs. 6 and 7 from 0 to 1 by 0.01 and adding all the piecewise distances between points in the feature space (See Figure 5 in Appendix). We want the distance ratio to be close to 1.0 for the embedding space distribution to be similar to that of the feature

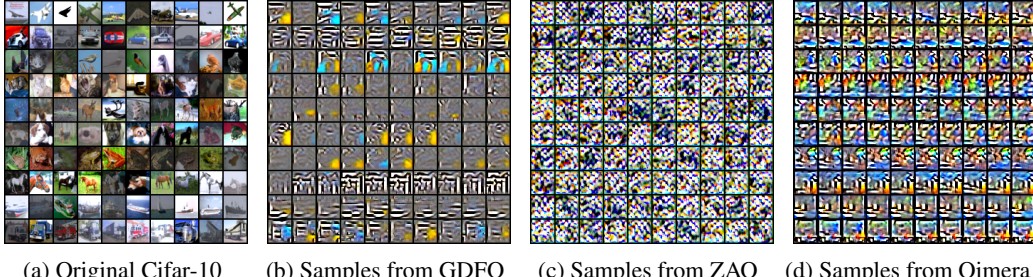

| (a) Original Cifar-10 | (b) Samples from GDFQ | (c) Samples from ZAQ | (d) Samples from Qimera |

Figure 4: Synthetic samples generated for Cifar-10 dataset. Each row represents one of the 10 classes, except for ZAQ which generates samples without labels.

space. Second, we defined 'intrusion score', the sum of logits that are outside the chosen pair of classes. If the score is large, that means the synthetic boundary samples are regarded as samples of non-related classes. Therefore, lower scores are desired. For comparison, we have also included the Mixup as a naive method and measured both metrics. It shows that the DM and EEI are effective for in terms of both the distance ratio and the intrusion score, explaining how DM and EEI achieves better performances.

# 6 Discussion

## 6.1 Does it Cause Invasion of Privacy?

The motivation for data-free quantization is that the private original data might not be available. Using a generator to reconstruct samples that follow the distribution of original data might indicate the invasion of privacy. However, as displayed in Figure 4, at least with current technologies, there is no sign of privacy invasion. The generated data are far from being human interpretable, which was also the case for previous works [15, 14, 13]. See Appendix (Figure 6) for more generated images.

Geirhos et al. [52] reveal an interesting property that CNNs are heavily biased by local textures, not global shapes. BagNet [53] supports this claim by confirming that images with distorted shapes but preserved textures can still mostly be correctly classified by CNNs. In such regard, it is no wonder that the generated samples are non-interpretable, because there is only a few combinations that maintains the global shape out of all possibilities that preserve the textures. Nonetheless, not being observed does not guarantee the privacy protection. We believe it is a subject for further investigations.

## 6.2 Limitations

Even though Qimera achieves a superior performance, one drawback of this approach is the fact that it utilizes embeddings of multiple classes to generate the boundary supporting samples. This restricts its application to classification tasks and its variants. For example, generation of datasets such as image segmentation or object detection do not take class embeddings as the input. However, generators for such data are not extensively studied yet, and we envision adding diversity to them would require inputting some form of labels or embeddings similar to paint-to-image [54], which would allow Qimera be easily applied.

# 7 Conclusion

In this paper, we have proposed Qimera, a simple yet powerful approach for data-free quantization by generating synthetic boundary supporting samples with superposed latent embeddings. We show in our motivational experiment that current state-of-the-art generative data-free quantization can be greatly improved by a small set of boundary supporting samples. Then, we show that superposing latent embeddings can close much of the gap. Extensive experimental results on various environments shows that Qimera achieves state-of-the-art performance for many networks and datasets, especially for large-scale datasets.

## Acknowledgments and Disclosure of Funding

This work was supported by National Research Foundation of Korea (NRF) grant funded by the Korea government(MSIT) (No.2021R1F1A1063670, No.2020R1F1A1074472) and Institute of Information & communications Technology Planning & Evaluation (IITP) grant funded by the Korea government (MSIT) (No.2020-0-01361, Artificial Intelligence Graduate School Program (Yonsei University))

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
