# Supplementary Materials for
# Qimera: Data-free Quantization with Synthetic Boundary Supporting Samples

## A   Code

The whole code is available at `https://github.com/iamkanghyunchoi/qimera`, including the training, evaluation, and visualization for all settings. This project code is licensed under the terms of the GNU General Public License v3.0.

## B   Additional Experimental Results

### B.1   Baseline with Different Noise Level

Instead of the superposed embeddings, we tried the alternative of adjusting the variance of the noise inputs, as discussed in Section 4.1 of the main body. We have tested five values as standard deviation $\sigma_z$: 0.25, 0.5, 1.0, 1.5, and 2.0. The results in Table 6 show that as we hypothesized, just by increasing the noise level did not provide much improvement on the performance of the baseline. Instead, we have noticed a slight improvement on GDFQ when the noise level is decreased, and we believe this is due to the generation of clearer samples. Nonetheless, the accuracy is far below that of the proposed Qimera.

| Dataset | Model | Bits | | GDFQ | | | | | Qimera |
|---|---|---|---|---|---|---|---|---|---|
| | | | $\sigma_z$ 0.25 | 0.50 | 1.0* | 1.5 | 2.0 | |
| Cifar-10 | ResNet-20 | 4w4a | 90.82 | 90.59 | 90.25 | 90.10 | 89.99 | 91.26 |
| | | 5w5a | 93.45 | 93.45 | 93.38 | 93.30 | 93.31 | 93.46 |
| Cifar-100 | ResNet-20 | 4w4a | 64.33 | 63.70 | 63.39 | 63.47 | 63.18 | 65.10 |
| | | 5w5a | 68.17 | 68.37 | 66.12 | 67.46 | 67.33 | 69.02 |
| ImageNet | ResNet-18 | 4w4a | 61.47 | 60.91 | 60.60 | 60.25 | 60.20 | 63.84 |
| | | 5w5a | 68.92 | 68.25 | 68.40 | 68.37 | 68.07 | 69.29 |
| | ResNet-50 | 4w4a | 55.30 | 54.37 | 52.12 | 54.29 | 46.14 | 66.25 |
| | | 5w5a | 72.01 | 72.56 | 71.89 | 71.60 | 71.14 | 75.32 |
| | MobileNetV2 | 4w4a | 60.29 | 59.90 | 59.43 | 58.67 | 59.31 | 61.62 |
| | | 5w5a | 68.69 | 68.35 | 68.11 | 68.10 | 67.84 | 70.45 |

*Default value from $\mathcal{N}(0, 1)$

Table 6: Experimental results on noise variance test

### B.2   Extracted Embedding Initialization without Training

To show that the weight from the last fully connected layer of the full-precision model is a good candidate for the initial embeddings, we have performed an experiment where the embeddings are

35th Conference on Neural Information Processing Systems (NeurIPS 2021).

| Dataset | Model (FP32 Acc.) | Bits | ZeroQ | ZAQ | GDFQ | Qimera | Extracted Init + Freeze |
|---|---|---|---|---|---|---|---|
| Cifar-10 | ResNet-20 | 4w4a | 79.30 | 92.13* | 90.25 | 91.26 | 90.37 |
| | (93.89) | 5w5a | 91.34 | 93.36 | 93.38* | 93.46 | 93.25 |
| Cifar-100 | ResNet-20 | 4w4a | 47.45 | 60.42 | 63.39* | 65.10 | 63.83 |
| | (70.33) | 5w5a | 65.61 | 68.70* | 66.12 | 69.02 | 68.76 |
| ImageNet | ResNet-18 | 4w4a | 22.58 | 52.64 | 60.60* | 63.84 | 63.67 |
| | (71.47) | 5w5a | 59.26 | 64.54 | 68.40* | 69.29 | 69.23 |
| | ResNet-50 | 4w4a | 8.38 | 53.02* | 52.12 | 66.25 | 63.20 |
| | (77.73) | 5w5a | 48.12 | 73.38* | 71.89 | 75.32 | 74.84 |
| | MobileNetV2 | 4w4a | 10.96 | 0.10$^{\dagger}$ | 59.43* | 61.62 | 60.46 |
| | (73.03) | 5w5a | 59.88 | 62.35 | 68.11* | 70.45 | 68.82 |

\* Highest among the baselines  $^{\dagger}$Did not converge

Table 7: Extracted embedding initialization without Training

frozen right after initialization. The results are presented in Table 7. Qimera with frozen embeddings are not better than the primary Qimera method with trained embeddings. However, compared to the two baselines (ZAQ and GDFQ), they provides a comparable accuracy on Cifar-10 and better accuracies on Cifar-100 and ImageNet. Furthermore, the accuracy on Cifar-10 dataset is close to the upper bound for all techniques under comparison, and thus the differences are minimal.

## B.3 Sensitivity Study on Number of DM layers

| Num. DM Layers | Accuracy | |
|---|---|---|
| | Cifar-10 | Cifar-100 |
| (w/o DM) | 90.81 | 64.89 |
| 1 | 91.26 | **65.10** |
| 2 | 91.18 | 64.90 |
| 4 | 91.49 | 64.96 |
| 8 | **91.63** | 64.11 |

Table 8: Sensitivity Study on Number of DM Layers

To have a deeper look into the DM layers, we have conducted a sensitivity study on the number of DM layers in Table 8. In the table, all results are from 4w4a setting with $p$=0.4, $K$=2 for Cifar-10 and $K$=10 for Cifar-100. As displayed, we found that there are sometimes small improvements from using more DM layers above one, but a severe drop in performance has been observed for using too many layers (Cifar-100, 8 layers).

## B.4 More Sensitivity Study on Hyperparameters

In addition to our choice of hyperparameters presented in the main body, we have performed a further extensive sensitivity study on those parameters, which is displayed in Table 9. All experiments are against 4w4a configuration, equal to the Table 3 (Section 5.4) in the main body. Regardless of the choice in $p$ and $K$, the results are all better than the two baselines ZAQ and GDFQ. Furthermore, while they all provide a meaningfully good performance, the results show a clear trend: lower $p, K$ for Cifar-10/100 and higher $p, K$ for ImageNet as sweet spots. This result supports the use of Qimera in that these parameters are easily tunable, not something that must be exhaustively seAutoReConhed for optimal values.

| Dataset | $K$ | $p$ | | | | | |
|---|---|---|---|---|---|---|---|
| | | 0.10 | 0.25 | 0.40 | 0.55 | 0.70 | 0.85 |
| Cifar-100 (ResNet-20) | 2 | 64.18 | 64.62 | 64.90 | 64.95 | 64.76 | 64.89 |
| | 10 | 64.85 | 64.63 | **65.10** | 64.76 | 64.52 | 63.86 |
| | 25 | 64.53 | 64.91 | 64.72 | 64.66 | 64.40 | 64.01 |
| | 100 | 64.37 | 64.66 | 64.64 | 64.27 | 64.79 | 63.65 |
| ImageNet (ResNet-50) | 100 | 58.74 | 60.64 | 61.43 | 61.47 | 63.87 | 65.73 |
| | 250 | 61.30 | 61.28 | 62.16 | 64.03 | 64.50 | 65.23 |
| | 500 | 58.96 | 60.11 | 58.69 | 63.05 | **66.25** | 66.19 |
| | 1000 | 58.65 | 59.62 | 61.20 | 58.86 | 65.12 | 64.24 |

Table 9: Further Sensitivity analysis

| Dataset | Model | Bits | DSG [1] | Qimera (%p improvement) |
|---|---|---|---|---|
| ImageNet | ResNet-18 | 4w4a | 34.53 | 63.84 (+29.31) |
| | ResNet-50 | 6w6a | 76.07 | 77.18 (+1.11) |
| | InceptionV3 | 4w4a | 34.89 | 73.31 (+38.42) |
| | SqueezeNext | 6w6a | 60.50 | 65.97 (+5.47) |
| | ShuffleNet | 6w6a | 44.88 | 56.16 (+11.28) |

Table 10: Comparison with DSG

### B.5 Comparison with DSG

Qimera is conceptually similar to DSG [1] which tries to diversify the sample generation by relaxing the batch-norm stat alignments. However, Qimera is different from DSG because we explicitly try to generate boundary supporting samples, instead of relying on diversification. This would led to better performance as demonstrated in the motivational experiment of Section 3.

Table 10 shows the comparison of Qimera with DSG. We use the reported numbers for DSG, and perform a new set of experiments for Qimera to match the settings. We use the lowest-bit settings for each network evaluated in DSG. As displayed in the table, Qimera outperforms DSG in all settings, especially for 4w4a cases.

## C   Class-Pairwise Visualization

To look closely onto the visualization of the samples from Section 5.3 (Figure 3) in the main body, we have plotted them in a pair-wise manner. Even though 10 classes in total gives 45 possible pairs, we chose nine symbolically adjacent pairs in the figure. Although being symbolically adjacent does not have much meaning, we believe having nine pairs is enough for our purpose rather than showing all 45 possible pairs. The colors match that of the Figure 3, where the lightgreen dots represent the synthetic boundary supporting samples. Also, we have plotted the path between the centroids of the two clusters in **black**, by varying $\lambda$ (the ratio of superposition) from 0 to 1 by 0.01 without any noise. Each 10th percentile is denoted as larger black dots. The results show that the samples and the path lie relatively in the middle of the two clusters. Please note that we have performed PCA plot for each pair to best show the distribution, so the position and orientation of the clusters do not exactly match those from Figure 3.

## D   More Generated Images

Lastly, Figure 6 shows more samples generated from Qimera. Figure 6a displays the synthetic boundary supporting samples generated from Cifar-10 dataset, with $K = 2$ and $\lambda = 0.5$. Each row and column represents a class from Cifar-10. For example, the image at row 0 (airplane) and column 2 (bird) represents a sample generated from superposed embeddings of airplane and bird. Although

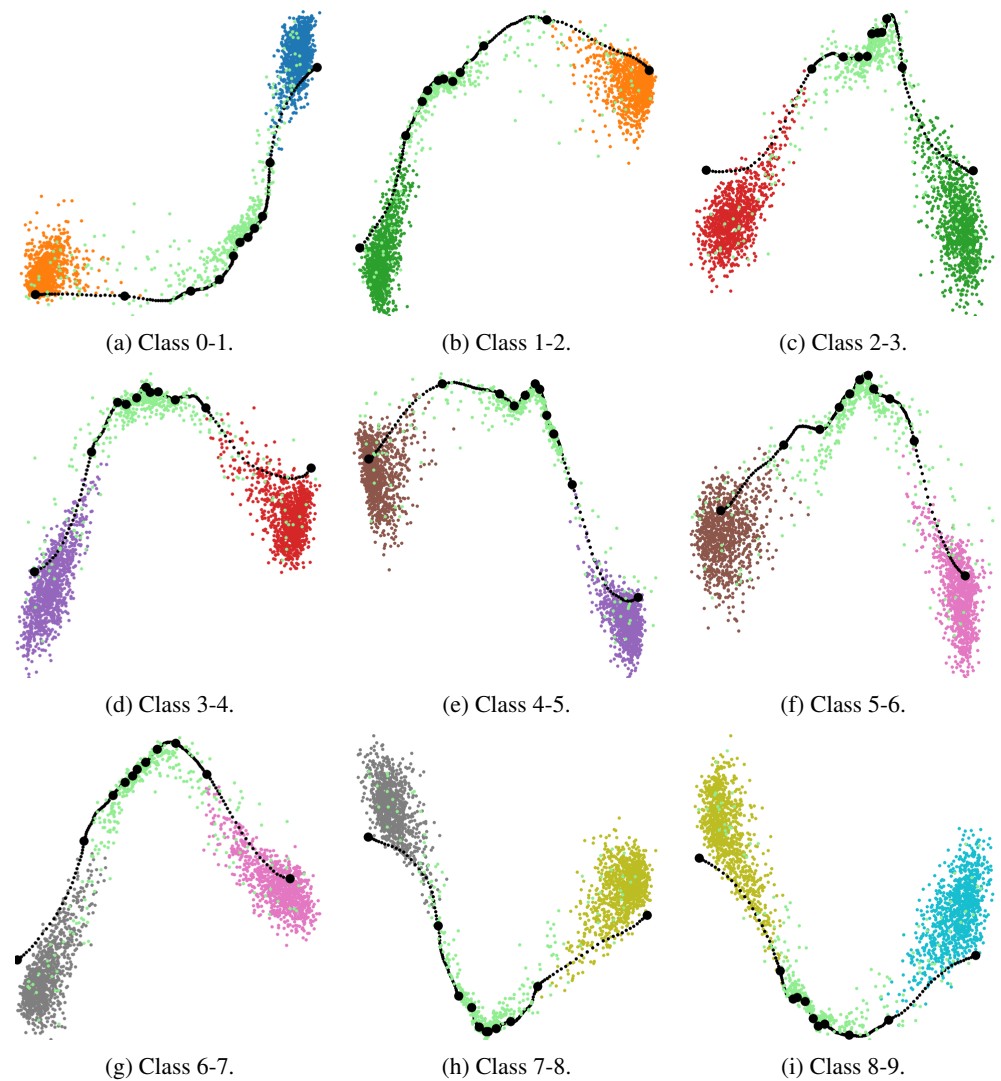

(a) Class 0-1.    (b) Class 1-2.    (c) Class 2-3.

(d) Class 3-4.    (e) Class 4-5.    (f) Class 5-6.

(g) Class 6-7.    (h) Class 7-8.    (i) Class 8-9.

Figure 5: Visualization of the generated samples in the feature space. The lightgreen cloud represents the synthetic boundary supporting samples. The black dots represents the path between the two embeddings without any noise, where every 10th percentile is denoted as larger dots. The colors match that of the Figure 3 of the main body, but the PCA dimension has been adjusted to best show each chosen class pair.

still not very human-recognizable, we find that each sample in Figure 6a has some features adopted from each of the source classes in Figure 4d.

Figure 6c shows the sample images created from ImageNet. Because there are too many classes within ImageNet (1000), we chose 10 classes from them, which are {0: 'tench, Tinca tinca', 100: 'black swan, Cygnus atratus', 200: 'Tibetan terrier, chrysanthemum dog', 300: 'tiger beetle', 400: 'academic gown, academic robe, judge's robe", 500: 'cliff dwelling', 600: 'hook, claw', 700: 'paper towel', 800: 'slot, one-armed bandit', 900: 'water tower'}, and the original samples from those classes are shown in Figure 6b. As in Cifar-10, the generated samples are far from human-recognizable, but each row is clearly distinguishable from the others. In addition, Figure 6d contains the synthetic boundary supporting samples from ImageNet, following the same rules from Figure 6a. Again, we see that each position in the sample matrix adopts features from the rows of the corresponding class pair in Figure 6c.

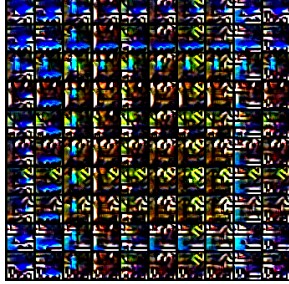

(a) Synthetic boundary supporting samples from Cifar-10.

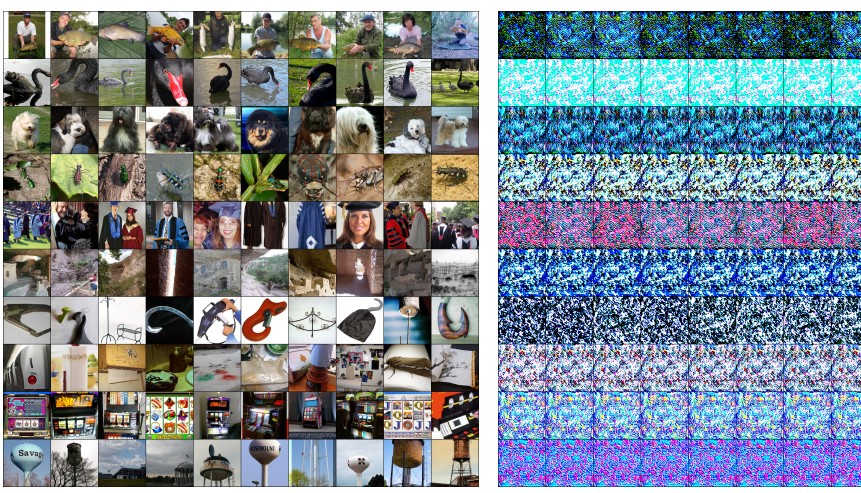

(b) Original ImageNet samples from the selected 10 classes.

(c) Synthetic samples from ImageNet.

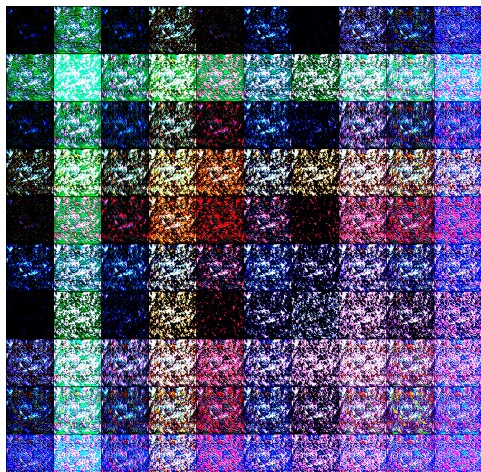

(d) Synthetic boundary supporting samples from ImageNet.

Figure 6: Additional synthetic samples.