# OpenReview forum: "Qimera: Data-free Quantization with Synthetic Boundary Supporting Samples"
_NeurIPS.cc/2021/Conference — NeurIPS 2021 Poster_

### Official Review · Reviewer_11Bt · 2021-07-13

**Rating:** 7
**Confidence:** 4

**Summary:**

In this paper, the authors proposed Qimera to improve data-free quantization by generating samples around the decision boundary. Qimera uses superposed latent embeddings for the generator to synthesize boundary supporting samples. To improve the superposition quality, the authors proposed a disentanglement mapping layer and extracted embedding initialization from the original fully connected layer to "flatten" the latent space. Experimental results show that Qimera outperforms existing work on data-free quantization.

**Limitations And Societal Impact:**

Yes

**Main Review:**

Pros:
1. Firstly, the paper is generally well written and easy to follow. The motivation experiments clearly show the necessity for boundary supporting samples.
2. The proposed techniques to flatten the latent space further improve the performance of latent space superposition.
3. Experimental results are solid. The ablation study on applying mixup to the baseline methods is very informative. It shows that interpolating in the latent space is a more effective regularization compared to interpolating in the image space.

Comments:
1. I am not very clear why the disentanglement mapping alone can help make the latent space more "flatten". Is it similar to the style mapping in the StyleGAN family? If so, does it help to use an MLP for the mapping?
2. Regarding the data leaking issue, in the Dreaming to Distill paper [25], the synthesized images are actually quite visually interpretable. Could you please also show the images on the ImageNet dataset? What is the difference that makes the synthesized images less interpretable in this work?
3. Regarding "use noise with higher variance" (L148), is it possible to just use KD loss for the generated samples with large noises and get rid of the incorrect label issue?


**Time Spent Reviewing:**

3

---

> ### Author Response · Authors · 2021-08-10
> **Response to Reviewer 11Bt**
>
> Thank you very much for the kind review. We respond to the reviewer’s concerns below:
> * *I am not very clear why the disentanglement mapping alone can help make the latent space more "flatten”*
>   * We apologize for the confusion. In the paper, we have used the term ‘flat’ to represent that changes to the embedding space are reflected in the feature space in a similar way. In the proposed method, superposed embeddings are supposed to help generate the boundary supporting samples. That is, when we superpose embeddings of two different classes, we want the features of the resulting samples to represent the boundary between the two classes. In such regard, if the representation is better disentangled, the superpositions of two or more embeddings have a better chance of correctly being located in the boundary between the corresponding features. We hope this clarifies, and will update the manuscript accordingly.
>
> * *Does it help to use MLP for the mapping?*
>   * We thank the reviewer for the valuable suggestion. Although we didn’t try it in the original submission, we conducted an additional experiment on the number of layers for the DM technique. The results are as below.
>
>
> |   **Dataset**  |  **DM Layers** | **Accuracy**    |
> |:-----------------|:--------------------:|:-------------------:|
> |  Cifar-10  |   (w/o DM)    |     90.81       |
> |  Cifar-10  |   1          |     91.26       |
> |  Cifar-10  |    2             |     91.18       |
> |  Cifar-10  |    4      |     91.49      |
> |  Cifar-10  |    8    |     91.63      |
> |  Cifar-100 |   (w/o DM)   |     64.89       |
> |  Cifar-100 |  1    |     65.10       |
> |  Cifar-100 |  2    |     64.90       |
> |  Cifar-100 |  4    |     64.96       |
> |  Cifar-100 |  8    |     64.11      |
>
>    In the table, all results are from 4w4a setting with $\rho$=0.4. K=2 for Cifar-10 and K=10 for Cifar-100. As displayed, we found that there are sometimes small improvements from using more DM layers above one, but a severe drop has been observed for using too many layers (Cifar-100, 8 layers). We will add the result to the final version of the paper.
>
>
> * *Dreaming to Distill paper [25], the synthesized images are actually quite visually interpretable. Could you please also show the images on the ImageNet dataset?*
>   - We included the ImageNet synthetic samples in the supplementary material. They also look like noises to humans. We believe the result from Dreaming to Distill is visually interpretable because they apply explicit loss for making neighboring pixels similar. We tried applying the same loss to our setting, but it resulted in achieving a lower performance.
>
> * *Regarding "use noise with higher variance" (L148), is it possible to just use KD loss for the generated samples with large noises and get rid of the incorrect label issue?*
>   * In fact, when we conducted the experiment in table 4 of the supplementary material, we applied KD loss to address the incorrect label issue with varioust weights on KD loss and the CE loss, including a case for 0 weight on the CE loss  (KD-only). As shown in the table, the performance was not noticeably better. We believe that although using KD-only loss can remove the wrong label problem, it still has the limitation of not having class-specific information.

---

> > ### Comment · Reviewer_11Bt · 2021-08-21
> > **Reply**
> >
> > Thanks for the informative response, which solves most of my concerns. After reading the rebuttal and the comments from other reviewers, I would like to stick to my acceptance recommendation.

---

### Official Review · Reviewer_84sZ · 2021-07-16

**Rating:** 6
**Confidence:** 5

**Summary:**

This paper addresses a popular data-free post-training quantization problem using a faux dataset of synthetically generated samples without depending on the BN layer statistics of the FP32 model. It proposes a superposed latent embeddings technique generating synthetic boundary samples to capture the distribution of the original data around the decision boundaries. The authors also introduce two auxiliary techniques to achieve additional performance gain.


**Ethical Concerns:**

No ethical concerns.

**Limitations And Societal Impact:**

The authors adequately addressed the limitations and no sign of privacy invasion.

**Main Review:**

This paper is well-written, and its goal is well-oriented by providing a good motivational experiment. The proposed Qimera approach provides a valuable insight that the synthetic samples of conventional data-free compression methods lack samples near the decision boundary of the FP32 model. The paper is well-organized and provides a unifying view of several generative data-free quantization techniques pioneered by ZeroQ.

The novelty is moderate but not strong enough because of the following reasons: Specifically, (1) a generative data-free quantization framework to generate a synthetic dataset has been studied in recent literature, say ZeroQ(CVPR'2020), GDFQ(ECCV'2020), ZAQ(CVPR'2021), and DSG(CVPR'2021). The proposed work seems to extend more like incremental modifications to the existing methods by combining already known concepts and ideas. Their philosophy and overall approaches are quite similar by following a generative data-free quantization framework originated from ZeroQ. Moreover, (2) the significance of its contribution appears limited in scope, as the boundary supporting samples have already been successful for knowledge distillation, especially in data-free network quantization with adversarial knowledge distillation(CVPR'2020) and knowledge distillation with adversarial samples supporting decision boundary(AAAI'2019). Therefore, the authors employed the main idea of generating boundary supporting samples to apply data-free quantization instead of knowledge distillation.

Experimental results look good. Still, its application is quite restricted to the classification task because of the inherent drawback of superposed latent embeddings to create samples of multiple classes, which is inferior to the other approaches like ZAQ and DSG extending the application to image segmentation and object detection beyond image classification.

Here are a few concerns and questions:
1. Please include the recently published DSG(Diversifying Sample Generation for Accurate Data-Free Quantization, CVPR'2021) paper as a reference. Comparison between the DSG and the proposed Qimera might be required. An additional evaluation comparing with DSG would be preferred.
2. Both ZAQ and DSG have improved the previous GDFQ without using BN statistics. Adding their PCA plots would be better to understand Qimera's superiority over them.
3. Is the amount of synthetic data generated by each benchmarking method all equal in the experiments conducted in this paper? It would be better to specify the amount of synthetic data used for fine-tuning in all methods.
4. For experiments with Cutmix and Mixup applied to GDFQ in Table 2, is it reasonable to apply these augmentation techniques to data focused on local textures as mentioned in the Discussion? Such augmentation techniques are thought to be efficient when global shapes are distinct. Therefore, it is inappropriate to conclude the performance degradation by applying Cutmix and Mixup to these synthetic data with a lack of boundary supporting samples. If appropriate, it would be better to visualize it or organize a separate experiment to prove it.
5. This paper experimentally demonstrates that synthetic samples generated near the decision boundary help the data-free quantized model's performance improve. The ablation study in Table 3 shows the performance changes of the model by Qimera's SE, DM, and EEI. The conceptual goal of SE seems to be on the opposite side of DM and EEI. Their combination would be quite strange because disentangling generally used in GAN is the opposite of superposing. The presented experimental results show the highest performance improvement is mainly due to the superposed latent embeddings. Does the ablation study setting still use the same amount of synthetic samples for SE, SE+DM, SE+EEI, and SE+DM+EEI? It would be good to provide valuable insights by carefully analyzing their performance difference because creating more synthetic samples could improve. Is there any other way to theoretically explain how the two auxiliary techniques improve using SE only? Or, if you visualize how DM and EEI can increase data near the decision boundary, you could see how boundary supporting samples improve the model's performance.


=================== post-rebuttal comment ===================

I thank the authors for their detailed response. The newly added experimental results address most of my concerns. Accordingly, I am happy to raise my rating.

===========================================================

**Time Spent Reviewing:**

40 hours

---

> ### Author Response · Authors · 2021-08-10
> **Response to Reviewer 84sZ**
>
> We would like to thank the reviewer for the thoughtful reviews that will help strengthen our paper. We would like to address the concerns as below:
>   * *Its application is quite restricted to the classification task because of the inherent drawback of superposed latent embeddings to create samples of multiple classes*
>
>     - As we discussed in the main body, we agree to the reviewer that this work is limited to problems with multiple classes. In fact, we are working on extending the work to non-classification problems (i.e., object detection and image segmentation) as our future work. We identify our contribution as achieving state-of-the-art performance on classification problems, which still forms an important field of application.
>
> * *Limited novelty, due to several prior papers on data-free quantization, and on boundary supporting samples.*
>   - The reviewer is right in that we are not the first to propose data-free quantization, or boundary-supporting samples, as those problems have been studied by much of the previous work. However, we would like to claim that our contribution is that we set a new SOTA for the existing data-free quantization problem, by adopting the idea of boundary supporting samples. Although boundary supporting samples have been previously used for knowledge distillation, we believe this is the first work to use it for data-free quantization.
>
> * *Please include the recently published DSG(Diversifying Sample Generation for Accurate Data-Free Quantization, CVPR'2021) paper as a reference.*
>   - We would like to thank the reviewer for pointing out an important piece of related work that we have missed out in the original submission. We will add DSG to the reference list. DSG addresses the diversity problem by allowing a certain degree of freedom on matching the batch-norm statistics. Although we could not replicate the work because the code is not yet available, we believe the work is orthogonal to ours as the idea of relaxed BN stat matching can also be applied to Qimera to improve performance.
>   - In the table below, we conducted additional experiments in the matching environment to compare our performance with the results published in the DSG paper. In all examined settings, Qimera outperforms DSG.
>
> |  **Dataset** |  **Model**        | **Precision** | **DSG** | **Qimera** |
> |:--------------|:----------------:|:------------:|:-----------------:|:--------------------:|
> |  ImageNet   | ResNet18 | 4w4a          | 34.53           | 63.84 (+29.31)  |
> | ImageNet    | ResNet50 | 6w6a          | 76.07           | 77.18 (+1.11)  |
> | ImageNet | InceptionV3 | 4w4a          | 34.89           | 73.31 (+38.42)  |
> |  ImageNet | SqueezeNext | 6w6a          | 60.50           | 65.97 (+5.47)
> |  ImageNet | ShuffleNet| 6w6a          | 44.88          | 56.16 (+11.28)  |
>
>
> * *Adding PCA for ZAQ and DSG would be better*
>
>   * Thank you for the suggestion. We added a PCA plot for ZAQ. As ZAQ does not produce samples based on class-wise embeddings, they tend to form a single large cluster instead of multiple per-class clusters. In addition, since DSG does not have a code available in public, we were unable to reproduce its result within the given time. We will try reproducing the paper and add its results as soon as possible.
>
> * *Is the amount of synthetic data generated by each benchmarking method all equal in the experiments conducted in this paper?*
>
>   * We apologize for the confusion. In all techniques, including the sensitivity/ablation study, we consistently fine-tune the quantized network for 200 iterations for 400 epochs. This fixes the number of synthetic samples, which is 5.12 million for Cifar-10/100 (200 * 400 * 64 batchsize) and 1.28 million (200 * 400 * 16 batchsize) for ImageNet. Therefore, it is not due to the increased number of synthetic samples that we are gaining the performance improvements. We will clarify this fact in the final version.
>
> * *Is it reasonable to apply Cutmix and Mixup to GDFQ?*
>
>   * We agree to the reviewer that Cutmix and Mixup are not appropriate methods for the synthetic samples. In fact, we were trying to make the same point the reviewer pointed out: Cutmix and Mixup are not suitable for data-free quantization. The reason we included those two methods is because they could be examples of naive solutions. If one were to try to make synthetic boundary samples from the baseline data-free quantization, we thought Cutmix and Mixup could be an easy pitfall that does not usually work. We apologize for the confusion, and thank you for pointing this out. We will clarify this in the final version.
>
> * *Combination of SE with DM and EEI seems odd because the conceptual goals are the opposite.*
>
>   * We apologize for the confusion. We agree to the reviewer that the disentanglement seems to be on the opposite side of superposed embeddings. However, the disentanglement is actually designed to help SE, rather than standing on the opposite side. In the proposed method, superposed embeddings are supposed to create boundary supporting samples. For example, when we superpose embeddings of two different classes, we want the features of the resulting samples to represent the boundary between the two classes. In such regard, if the representation is better disentangled, the superpositions of two or more embeddings have a better chance of correctly being located in the boundary between the corresponding features. We hope this clarifies, and will update the paper for clarification.
>
> * *Analyze the performance improvements coming from using different components of the proposed techniques (SE, SE+DM, SE+EEI, and SE+DM+EEI) and using more synthetic samples.*
>   * As we clarified above, we use an equal number of synthetic samples for all techniques (Please see above), and thus the performance improvements are from each method that is used, and not the increased number of synthetic samples. We apologize again for the confusion, and we will update the manuscript.
>
> * *Is there any other way to analyze the two auxiliary techniques (DM and EEI)?*
>   * Thanks for the suggestion. To further analyze DM and EEI, we investigated how DM and EEI help shaping the embedding space friendly to SE by measuring two metrics. First, setting K=2 (between two classes), we measured the ratio between the perceptual distance (length of the paths in Fig. 5) divided by Euclidean distance between the two embeddings in the classifier’s feature space. We want the ratio to be close to 1.0 for the embedding space distribution to be similar to that of the feature space. Second, we defined "intrusion score", the sum of logits that are outside the chosen pair of classes. If the score is large, that means the synthetic boundary samples are regarded as samples of non-related classes. Therefore, lower scores are desired. For comparison, we have also included the Mixup as a naive method and measured both metrics. Results are as follows:
>
>    |**Dataset**   |      **Method**     | **Dist.Ratio** |   **Intrusion**|
> |:-----------------|:--------------------|:-------------------:|:----------------:|
> |  Cifar-10   |  Mixup             |    2.44       |      0.800|
> |  Cifar-10   |  SE Only;         |    1.58       |      0.00260|
> |  Cifar-10   |  SE+DM                   |    1.67       |      0.00073|
> |  Cifar-10   |  SE+DM+EEI  |    1.57       |      0.00013|
> |  Cifar-100  |  Mixup            |    3.14       |      0.400|
> |  Cifar-100  |  SE Only        |    1.64       |      0.053|
> |  Cifar-100  |  SE+DM         |    1.59       |      0.044|
> |  Cifar-100  |  SE+DM+EEI |    1.52       |      0.029|
>
>   * It shows that the DM and EEI are effective for making a flat representation for the embedding space, making it easier for SE to improve performance. We will add this data in the final version of the paper.

---

### Official Review · Reviewer_syC3 · 2021-07-19

**Rating:** 6
**Confidence:** 4

**Summary:**

In the manuscript, the authors propose Qimera, which introduces superposed latent embeddings, disentanglement mapping, and extracted embedding initialization to generate boundary supporting samples.
These synthetic samples are useful for post-training data-free quantization.
The main contributions of the manuscript are:
1) Pioneer at considering boundary supporting samples for data-free quantization,
2) Propose Qimera and its components,
3) Achieve decent experimental results, together with meaningful analysis.


**Limitations And Societal Impact:**

The potential negative societal impact of this work is regarding data privacy.
Data-free quantization aims to quantize pretrained neural networks without access to the data they are trained on.
This generally requires an approximation of the distribution of training data derived/distilled from the neural networks.
The manuscript explicitly discuss this impact and show many results sampled in the approximated distribution of training data.
Since most samples are not interpretable by humans, the proposed method is good to use (suppose that the texture of training data is not private and interpretable).

**Main Review:**

I think the paper is generally interesting, and I may change my rate according to the rebuttal.
Strength:
1) The writing of this manuscript is good, with sufficient background knowledge, clear equations and explanations. Although the Qimera method has lots of components, the manuscript is still easy to follow.
2) The related work section is thorough.
3) Illustrations such as Figure 3 are helpful to validate the effectiveness of proposed methods in generating boundary supporting samples.
4) The experimental results are quite good comparing with previous methods targeting using synthetic data on post-training quantization (PTQ).

Suggestion:
The supplementary materials contain lots of additional experiments and ablation study, which are helpful to address a large portion of my concerns (for example, the sensitivity to other hyperparameters, comparing with more recent methods, etc). I suggest merging some of them into the main texts if possible.

Weakness and Question:
1) Regarding the disentanglement mapping, M is implicitly encouraged to map the inputs to a flatter space because we suppose unnatural samples can lead to a large loss. Some ablation study or statistics are required to validate the point.
2) Qimera is good at generating synthetic images for PTQ. But will a better decision boundary be learned when the network has better effective capability (when applied with advanced quantization methods such as better quantizer or mixed-precision quantization)? Can Qimera improve on top of SOTA quantization methods? (ZAQ + Qimera experiments in supplementary material is a start, but ZAQ is not specifically targeting quantizers/precisions, and the ZAQ + Qimera results are not super promising).
3) Why the distributions of the embeddings should be similar to the feature space in order to be flat?
4) In addition to PTQ, it would be interesting to see if Qimera can help quantization-aware training (QAT) with synthetic data?
5) From the ablation, solely applying SE can have a good improvement, but applying mixup on inputs doesn't work well. It would be good to see some experimental results on the source of contributions. Is embedding the only key factor?

=========Post Rebuttal=========

Thanks the authors for the detailed rebuttal, which addressed many of my concerns. Based on the rebuttal and reviews from other reviewers, I will keep my rating and would like the paper to be accepted to NeurIPS 2021.

**Time Spent Reviewing:**

8

---

> ### Author Response · Authors · 2021-08-10
> **Response to Reviewer syC3**
>
> We appreciate the constructive review. We will address the reviewer’s concerns as shown below.
>
> * *Moving some data in supplementary materials to the main text is suggested*
>
>   - Thanks for the suggestion. In the original submission, we selected a few results that we thought were essential. However, the review pointed out that the sensitivity study and the comparison with existing methods are important enough to be placed in the main document. Should the paper be accepted, we plan to merge table 6 (further sensitivity), 7 (Qimera+ZAQ, with the updates. please see below), 8 (Comparison With AutoReCon) with the corresponding discussion to the main body by squeezing some text throughout the paper. Please kindly advise if there are further ideas.
>
>
> * *Can Qimera improve based on SOTA algorithms?*
>
>   - Thank you for the question. We believe Qimera is orthogonal to most of the generative quantization methods and could be applied to improve their performance. For example, We believe DSG (CVPR 2021) can be applied with our method in conjunction. We chose GDFQ due to two reasons. First, when using the official implementations from the authors, GDFQ was better than ZAQ in many settings, as shown in Table 2. Therefore, we believe GDFQ represents SOTA, especially in terms of stably achieving good performance. In addition, ZAQ is an exceptional scheme as they use adversarial training, which is well-known to be sensitive and has led to divergence in some of our experiments.
> Instead of using the official code, we tried tuning the hyperparameters of Qimera+ZAQ, and we were able to improve the performance as follows:
>
> |**Dataset** | **Model** | **Precision** | **ZAQ**  | **Qimera+ZAQ**|
> |--------------|--------------|:-----------:|-----------|------------------------|
> | Cifar-10 | ResNet20 | 4w4a        | 92.13 | 93.91 ± 0.06 (+1.78) |
> | Cifar-10 | ResNet20 | 5w5a        | 93.36 | 93.84 ± 0.06 (+0.47) |
> | Cifar-100|ResNet20 | 4w4a        | 60.42 | 69.30 ± 0.42 (+8.88) |
> | Cifar-100|ResNet20 | 5w5a        | 68.70 | 69.58 ± 0.15 (+0.88) |
>
>
>
>    - In addition, we have implemented Qimera on top of AutoReCon, and the results are as follows:
>
> |  **Dataset** |  **Model**        | **Precision** | **AutoReCon** | **Qimera+AutoReCon** |
> |--------------|:----------------:|:------------:|:-----------------:|:--------------------:|
> |  Cifar-10   | ResNet20 | 4w4a          | 88.55           | 91.16 (+2.61)  |
> |  Cifar-10   | ResNet20 | 5w5a          | 92.88           | 93.42 (+0.54)  |
> |  Cifar-100 | ResNet20 | 4w4a          | 62.76           | 65.33 (+2.57)  |
> |  Cifar-100 | ResNet20 | 5w5a          | 68.40           | 68.80 (+0.40)  |
>
>
>   The result shows that Qimera can be built based on various methods, including the recent SOTA to improve performance.
>
> * *Why should the distributions of the embeddings be similar to the feature space in order to be flat?*
>   - We apologize for the confusion. In the paper, we used the term ‘flat’ to represent that the changes to the embedding space are reflected in the feature space in a similar way. In the proposed method, superposed embeddings are supposed to help generate boundary supporting samples. For example, when we superpose embeddings from two different classes, we want the features of the resulting samples to represent the boundary between the two classes. Although there could be multiple ways of achieving this, an effective way would be having a similar distribution for the embedding space and the feature space, which is what we tried to achieve.
>
> * *Some ablation or sensitivity study of DM is needed*
>    - Thanks for the suggestion. We performed two additional experiments to study the behavior of the DM layers, and they provide more insights. We will update the manuscript with these results.
> We conducted a sensitivity study for the number of DM layers, and the results are as below:
>
> |   **Dataset**  |  **DM Layers** | **Accuracy**    |
> |:-----------------|:--------------------:|:-------------------:|
> |  Cifar-10  |   (w/o DM)    |     90.81       |
> |  Cifar-10  |   1          |     91.26       |
> |  Cifar-10  |    2             |     91.18       |
> |  Cifar-10  |    4      |     91.49      |
> |  Cifar-10  |    8    |     91.63      |
> |  Cifar-100 |   (w/o DM)   |     64.89       |
> |  Cifar-100 |  1    |     65.10       |
> |  Cifar-100 |  2    |     64.90       |
> |  Cifar-100 |  4    |     64.96       |
> |  Cifar-100 |  8    |     64.11      |
>
>   In the table, all results are from 4w4a setting with $\rho$=0.4. K=2 for Cifar-10 and K=10 for Cifar-100. As displayed, we found that there are sometimes small improvements from using more DM layers above one, but a severe drop in performance has been observed for using too many layers (Cifar-100, 8 layers). We will add the result to the final version of the paper.
>
>
>   - We investigated how DM and EEI help shaping the embedding space friendly to SE. First, setting K=2 (between two classes), we measured the ratio between the perceptual distance (length of the paths in Fig. 5) divided by Euclidean distance between the two embeddings in the classifier’s feature space. We want the ratio to be close to 1.0 for the embedding space distribution to be similar to that of the feature space. Second, we defined `intrusion score’, the sum of logits that are outside the chosen pair of classes. If the score is large, that means the synthetic boundary samples are regarded as samples of non-related classes. Therefore, lower scores are desired. For comparison, we have also included the Mixup as a naive method and measured both metrics. Results are as follows:
>
>   |**Dataset**   |      **Method**     | **Dist.Ratio** |   **Intrusion**|
> |:-----------------|:--------------------|:-------------------:|:----------------:|
> |  Cifar-10   |  Mixup             |    2.44       |      0.800|
> |  Cifar-10   |  SE Only;         |    1.58       |      0.00260|
> |  Cifar-10   |  SE+DM                   |    1.67       |      0.00073|
> |  Cifar-10   |  SE+DM+EEI  |    1.57       |      0.00013|
> |  Cifar-100  |  Mixup            |    3.14       |      0.400|
> |  Cifar-100  |  SE Only        |    1.64       |      0.053|
> |  Cifar-100  |  SE+DM         |    1.59       |      0.044|
> |  Cifar-100  |  SE+DM+EEI |    1.52       |      0.029|
>
>   It shows that the DM and EEI are effective for making a flat representation for the embedding space, making it easier for SE to improve performance. We will add this data in the final version of the paper.
>
> * *It would be interesting to see the source of SE vs mixup*
>   - Thanks for the inspiring question. To address this issue, we performed two separate experiments. First, on top of Figure 5, we plotted the features of mixup samples. Although we cannot present the images here, it clearly shows that the mixup samples are placed on arbitrary locations in the PCA plot, being a piece of evidence that the mixup samples are not adequate for synthetic BSS. Second, from the result in the table from the above item, Mixup is clearly inferior to SE in terms of both distance ratio and intrusion score. This shows that mixup samples do not represent the boundary between classes, and they are often regarded as samples of non-related classes.
>
>
> * *It would be interesting to see if Qimera can help QAT*
>   - Thank you for an excellent suggestion. Although we cannot demonstrate the idea on this rebuttal due to insufficient time, we believe the idea used in Qimera can be applied to QAT schemes in some settings and would make a great future work. Qimera can be used when the training methods use generators. We can think of the following few scenarios of QAT: 1) Adversarial robustness. Quantized networks are known to be vulnerable to adversarial attacks [a1,a2]. One popular scheme to mitigate this issue is to train against several generated adversarial samples. 2) Few-shot domain adaptation: When a quantized network is to be adapted to another domain, generative methods are sometimes used to regenerate original images [a3]. In such cases, we believe Qimera can help the adaptation process. 3) Continual learning: Continual learning is a problem under active research, and quantization would make the problem even more complicated. For continual learning approaches that use generators [a4], Qimera could help improve the performance of such problems.
>
>   [a1] Lin+, Defensive Quantization: When Efficiency Meets Robustness (ICLR, 2018)
>
>   [a2] Song+, Improving Adversarial Robustness in Weight-quantized Neural Networks (arxiv, 2020)
>
>   [a3] Sankaranarayanan+, Generate to Adapt: Aligning Domains Using Generative Adversarial Networks (CVPR 2018)
>
>   [a4] Shin+, Continual Learning with Deep Generative Replay  (NeurIPS 2017)

---

### Author Response · Authors · 2021-08-10
**General Response**

We thank all reviewers for their time and thoughtful comments. We have replied to each review and addressed their concerns. To list a few highlights,
* Conducted a few more experiments for comparing our work against DSG (CVPR 2021) and adding additional results for Qimera implemented on top of ZAQ (CVPR 2021) and AutoReCon (IJCAI 2021).
* Performed an additional sensitivity study to further analyze our two auxiliary techniques, DM and EEI.
* Clarified the intent for DM and EEI, which is also backed by the aforementioned experimental analysis

---

### Decision · Program_Chairs · 2021-09-27

**Decision:**

Accept (Poster)

**Comment:**

This paper addresses post-training data-free quantization to compress neural networks, an important domain of NN research. The reviewers found the approach, generating samples around the NN's decision boundaries (boundary supporting samples), and associated methods well motivated (though there were some clarification requests in the reviews, which the authors fulfilled). There were some concerns about similarities to prior work, both in motivations and methods: the authors agreed to include these and I encourage the authors to be generous in giving motivational credit to these works. The authors also provide additional experimental results in the rebuttal, which further demonstrated the effectiveness in the method. Therefore I recommend acceptance as a poster.